# Overcoming Ibrutinib Resistance in Chronic Lymphocytic Leukemia

**DOI:** 10.3390/cancers11121834

**Published:** 2019-11-21

**Authors:** Bartosz Puła, Aleksandra Gołos, Patryk Górniak, Krzysztof Jamroziak

**Affiliations:** 1Department of Hematology, Institute of Hematology and Transfusion Medicine, 02-776 Warsaw, Poland; kjamroziak@ihit.waw.pl; 2Institute of Hematology and Transfusion Medicine, 02-776 Warsaw, Poland; agolos@ihit.waw.pl; 3Department of Experimental Hematology, Institute of Hematology and Transfusion Medicine, 02-776 Warsaw, Poland; pgorniak@ihit.waw.pl

**Keywords:** Bruton’s tyrosine kinase, ibrutinib, chronic lymphocytic leukemia, treatment, resistance

## Abstract

Ibrutinib is the first Bruton’s tyrosine kinase (BTK) inhibitor, which showed significant clinical activity in chronic lymphocytic leukemia (CLL) and small lymphocytic lymphoma (SLL) patients regardless of cytogenetic risk factors. Recent results of phase III clinical trials in treatment-naïve CLL patients shift the importance of the agent to frontline therapy. Nevertheless, beside its clinical efficacy, ibrutinib possesses some off-target activity resulting in ibrutinib-characteristic adverse events including bleeding diathesis and arrhythmias. Furthermore, acquired and primary resistance to the drug have been described. As the use of ibrutinib in clinical practice increases, the problem of resistance is becoming apparent, and new methods of overcoming this clinical problem arise. In this review, we summarize the mechanisms of BTK inhibitors’ resistance and discuss the post-ibrutinib treatment options.

## 1. Introduction

Chronic lymphocytic leukemia (CLL) is an incurable clonal proliferation of small CD5/CD19-positive lymphocytes accumulating in blood, bone marrow and lymphoid tissues accounting for approximately 25% of all leukemias in Europe and North America [1,2]. The median age of diagnosis is 70 years, and the disease is incurable in majority of cases [3,4]. The disease has a heterogenous clinical course, and numerous studies aimed at identifying novel prognostic and predictive factors are being intensively performed. Although immunochemotherapy tailored to patients’ fitness has so far remained the backbone of frontline CLL treatment in the majority of patients, the development of novel selective compounds targeting the B cell receptor (BCR), along with increasing knowledge of predictive factors, have improved patients’ prognosis [3,4]. Next generation sequencing (NGS) identified a number of mutations in genes related to regulation of key cellular processes, e.g., response to DNA damage and cell cycle control (*ATM*, *TP53*, *RB1*, *BIRC3*), RNA processing (*SF3B1*), Notch signaling (*NOTCH1*, *NOTCH2*, *FBXW7*) and cytokine signaling (*NRAS*, *KRAS*, *BRAF*, *MYD88*, *DDX3X*, *MAPK1*) that modify CLL’s clinical course [5,6,7,8,9]. Although some of the above genetic changes were identified to yield prognostic impact, currently only the negative prognostic status of the p53 pathway aberrations is reflected in clinical guidelines [3,4]. Patients characterized by p53 protein defects (short arm of chromosome 17 deletion [del(17p)] or *TP53* gene mutation) are refractory or achieve only transient responses to anti-CD20 antibody based immunochemotherapy [10,11]. Furthermore, transformation to poor prognostically aggressive diffuse large B-cell lymphoma (DLBCL) also occurs in up to 10% of cases [12].

The outcome of CLL patients who are refractory or relapsed to immunochemotherapy changed with the development of novel agents inhibiting BCR signaling, e.g., Bruton’s tyrosine kinase (BTK) inhibitor ibrutinib and the phosphoinositide 3-kinase (PI3K) delta inhibitor idelalisib [4,13,14]. Both compounds presented remarkably high activity in CLL, including patients with p53 dysfunction [13,14,15]. Significant clinical efficacy of ibrutinib along with good tolerability, also in comorbid patients, were reported for both relapsed/refractory (RR-CLL) and treatment-naïve CLL (TN-CLL) [16,17]. Considering the widespread use of ibrutinib and other BTK inhibitors (BTKi) in current clinical practice, in this work we discuss the mechanism of action of BCR and ibrutinib in normal and pathological cells, and the adverse event profile of the drug. Furthermore, we present the most important findings regarding the resistance mechanisms to ibrutinib, reasons of therapy discontinuation, and put special emphasis on potential strategies and alternative compounds with the potential to overcome these clinical issues. 

## 2. B Cell Receptor Signaling in Normal and Pathological Cells

The cellular origin of B-cell lymphomas has been extensively studied over past 15 years. Early studies using gene expression profiling showed that malignant B cells originate from normal B-cells at a different stage of maturation [18,19,20,21]. Every normal B cell, and consequently every lymphoma cell, has a unique BCR consisting of pairs of immunoglobulin heavy (IgH) and light (IgL) chains. Each IgH and IgL has a unique variable (V) region that allows the BCR to bind to diverse antigens. The antibody portion of BCR is coupled on cell membranes with CD79A and CD79B subunits which mediate signal transductions [21].

In normal and lymphoma B cells, there are two modes of signaling involving the BCR: the antigen-independent “tonic” signaling and antigen-dependent “active” BCR signaling. Tonic BCR signaling was defined by the observation that the conditional ablation of surface BCR expression in mouse B-cells results in the eventual loss of all naive mature B-cells [22,23]. Tonic BCR signaling requires the immunoreceptor tyrosine-based activation motif (ITAM) portion of CD79A, but may not require the extracellular portions of IgM, suggesting that this mode of BCR signaling is antigen-independent [23,24]. A constitutively active form of the PI3K was able to rescue the survival of mouse B-cells in which the BCR was genetically ablated, suggesting a key role for PI3K in delivering survival signaling during tonic BCR signaling [25]. In contrast, active BCR signaling occurs subsequent to BCR aggregation, allowing SRC family kinases to phosphorylate CD79A, CD79B and spleen tyrosine kinase (SYK), which, in turn, activates BTK, PI3Kδ and the phospholipase C gamma 2 (PLCγ2). Unlike tonic BCR signaling, active BCR signaling engages many pathways and transcriptional networks that include the PI3K, mitogen-activated protein kinase (MAPK), nuclear factor of activated T cells (NFAT), RAS pathways and CARD11-mediated activation of NF-κB. Increased activity of NF-κB is characteristic of this mode of BCR signaling, which promotes proliferation and survival of normal and malignant B-cells [21,26]. 

Microscopic examination of the BCR on the surface of activated B cell type diffuse large B-cell lymphoma (ABC-DLBCL) cell lines and primary tumor cells revealed a consistent pattern of BCR clustering reminiscent of BCR clusters observed in antigen-stimulated normal B cells [26,27].

Moreover, it was shown that in approximately 30% of patients with CLL, BCRs have specific, almost identical structures that maybe classified into distinct subsets (defined as ‘stereotyped BCRs’) on the basis of shared sequence motifs within the *IGHV* genes (that is, *IGHV*–*IGHD*–*IGHJ* gene rearrangement sequences) [28]. The reactivity of BCRs to autoantigens exposed on apoptotic cells has been reported for CLL and ABC-DLBCL [29,30]. By expressing CLL-derived or lymphoma-derived BCRs in cell lines, investigators demonstrated that malignant BCRs bound self-antigens, which included structural elements within a subdivision of the immunoglobulin heavy chain V region known as the framework region (FR), triggering proliferation and survival signals in a cell-autonomous fashion [31,32]. Aside from autoantigens, it was shown that BCRs on CLL cells also respond to foreign antigens of bacteria and fungi which, as shown in mouse models, may stimulate CLL pathogenesis due to induced cross-reactivity with autoantigens [33,34,35].

Detailed understandings of the fundamental pathogenetic role of BCR signaling in B-cell lymphomas have led to the development of clinical modulators of this pathway, e.g., BTK, PI3K and SYK inhibitors (Figure 1). 

Bruton’s tyrosine kinase is a Tec protein tyrosine kinase (TEC)-family nonreceptor tyrosine kinase that signals downstream of numerous cellular receptors, including the BCR, toll-like receptors (TLR) and Fc receptors [36]. BTK transduces signaling downstream of the BCR and activates PLCγ2, which catalyzes the cleavage of membrane phosphatidyl-inositol 4,5 bisphosphate (PIP_2_) into inositol triphosphate (IP_3_) and diacylglycerol (DAG). This mobilizes calcium and activates protein kinase C beta (PKCβ) and downstream proteins. In addition, the BCR co-receptor transmembrane protein CD19 is phosphorylated by LYN during BCR signaling. This recruits PIK3 to BCR with subsequent phosphorylation of PIP_2_ to generate phosphatidylinositol-3,4,5-trisphosphate (PIP_3_). Collectively, these signaling pathways induce activation of the NF-κB, AKT, RAS, mitogen-activated protein kinase and nuclear factor of activated T cell pathways, resulting in B-cell cellular changes including cell survival, proliferation, adhesion, migration and homing [37,38,39,40]. Recently, Phelan et al. discovered a new mode of oncogenic BCR signaling in ibrutinib-responsive cell lines and DLBCL biopsies, coordinated by a multiprotein super complex formed by MYD88, TLR9 and BCR (My–T–BCR super complex). The My–T–BCR super complex co-localizes with the CBM complex, phosphorylated IκBα and mTOR on endolysosomes, where it drives pro-survival NF-κB and mTOR signaling. The My–T–BCR super complex was reduced by ibrutinib, but was further attenuated by the addition of AZD2014, an mTOR inhibitor. This dual mTOR and BTK inhibition cooperatively decreased MYD88 protein abundance and mTOR activity, providing mechanistic insights into synergism between BTK and mTOR or PI3K inhibitors. Additionally, My–T–BCR super complexes characterized ibrutinib-responsive malignancies and distinguished ibrutinib responders from non-responders [41]. Although My–T–BCR super complexes were not evident in CLL [41], these data shed more light on the inhibiting properties and resistance mechanisms of ibrutinib that should be investigated in near future.

Since BTK transduces constitutive signaling downstream of the BCR in many B cell malignancies, the protein has long been considered an attractive therapy target. Ibrutinib, a covalent BTKi has been approved for use in patients with CLL, Waldenström macroglobulinemia (MW), mantle cell lymphoma (MCL) and marginal zone lymphoma (MZL) [42].

## 3. Ibrutinib Mechanism of Action in CLL

In CLL cells, BTK is constitutively active and expressed at higher levels as compared to normal B cells warranting their proliferation and survival [43,44]. Ibrutinib (PCI-32765) is an orally bioavailable irreversible inhibitor of BTK binding covalently to sulfhydryl C481 residue of active BTK form inhibiting therefore further signal transduction leading to loss of proliferation, induction of apoptosis and cell activation [43,44]. Furthermore, ibrutinib diminishes intracellular interactions of CLL cells with surrounding microenvironment cells despite synthesis of tumor promoting cytokines such as B cell activating factor (BAFF), tumor necrosis factor alpha (TNFα) or CD40 ligand [43,44]. As BTK acts also downstream of chemokine receptors (CXCR4, CXCR5) and administration of ibrutinib leads to diminished cell adhesion resulting in promotion of CLL cells migration to peripheral blood with concomitant reduction of lymphatic organ infiltration [45,46]. CLL cells deprived of protective microenvironmental signals of lymphatic nurse-like cells undergo apoptosis; however, the rate is modest, and persistent lymphocytosis in CLL patients under ibrutinib therapy may be observed [47]. 

Ibrutinib is relatively selective towards BTK; however, the compound has some off-target activity such as the inhibition of several other tyrosine kinases and their pathways, e.g., epidermal growth factor receptor (EGFR), interleukin-2-inducible T cell kinase (ITK), T cell X chromosome kinase (TXK) and PI3K, owing to its efficacy on the one hand, but potentially contributing to a specific toxicity profile, as discussed below, on the other [48].

## 4. Clinical Activity of Ibrutinib

In the phase I study (#NCT00849654), ibrutinib monotherapy showed a 69% overall response rate (ORR) with 16% patients achieving complete response (CR) [49]. In the multicenter phase Ib-II trial (#NCT01105247), 85 RR-CLL patients received either 420 mg daily (n = 51) or 840 mg (n = 34) ibrutinib in monotherapy leading to 71% ORR, with only two patients showing CR in the 420 mg cohort [50]. With no significant differences in treatment efficacy and tolerability, the 420 mg daily dose was selected for further clinical trials. The RESONATE trial (#NCT01578707) was the first one to present superiority of ibrutinib monotherapy over monotherapy with ofatumumab (an anti-CD20 antibody) in relation to ORR (63% vs. 4.1%), progression-free survival (PFS) and overall survival (OS) [13]. Of note, ibrutinib was capable of diminishing the negative prognostic impact of 17p deletion. The recently published update of the study showed that ibrutinib may be safely administered for longer periods of time [51]. Importantly, ORR to ibrutinib increased over time, with 91% of patients obtaining a response at a median duration of ibrutinib treatment of 41 months. At data cut-off, 46% of patients continued treatment at a median follow-up of 44 months. Analysis of prognostic factors showed that ibrutinib activity was not influenced by baseline risk factors; however, patients with more than two prior therapies or with a presence of *TP53* and *SF3B1* mutations had a trend towards shorter PFS [51].

Ibrutinib monotherapy showed even better results in TN-CLL patients. The randomized phase III trial RESONATE II (#NCT01722487) revealed its superiority over chlorambucil monotherapy in terms of ORR, PFS and OS [16]. Although both RESONATE and RESONATE II trials confirmed the remarkable clinical efficacy of ibrutinib monotherapy, the above-mentioned trials were criticized for the agents used in the comparator arms as these were not up to date with immunotherapy regimens recommended by clinical guidelines [17,52,53,54]. The results of the recently published phase III clinical trials showed that ibrutinib based regimens produced a better clinical benefit than currently recommended first line immunochemotherapy regimens [17,53,54]. In the E1912 trial (#NCT02048813), the ibrutinib–rituximab regimen was characterized by longer PFS and OS than that observed with the use of FCR regimen in TN-CLL patients 70 years of age or younger [53]. Ibrutinib alone or combined with rituximab was also superior in terms ORR and PFS than the BR regimen in TN-CLL patients aged 65 years or more (#NCT01886872) [17]. Furthermore, in their study, Woyach et al. found that addition to rituximab, ibrutinib does not influence patient outcome [17]. Ibrutinib-rituximab was also superior in terms of ORR and PFS compared to an obinutuzumab-chlorambucil regimen in TN-CLL patients either aged 65 years or older, or younger than 65 years with coexisting medical conditions (#NCT02264574) [54]. Of note, the latter two trials of Woyach et al. and Moreno et al., due to a possible crossover to ibrutinib based regimens, were not capable of presenting differences in OS between cohorts [17,54].

## 5. Ibrutinib Adverse Events

Clinical trials and real-world data on ibrutinib treatment in frontline and relapse-refractory settings in CLL/SLL patients showed a favorable safety profile [4,13,15,16,17,51,53,54,55,56,57,58,59,60,61,62]. The prevailing ibrutinib-related adverse events included diarrhea (up to 60%), arthralgia (up to 40%), rash (up to 25%), infections mainly of the upper respiratory tract (up to 35%) and mild hematological toxicity occurred in up to 60% of patients. The most common grade 3 and higher adverse events comprised of infections (up to 30%) and hypertension (up to 25%); however, grade 3 and higher neutropenia, anemia and thrombocytopenia occurred rarely (up to 15%, 10% and 10% respectively). 

Although ibrutinib treatment was well tolerated, some unexpected adverse events, probably due to off-target activity, have also been reported and included mainly atrial fibrillation and hemorrhagic complications [4,13,15,16,17,51,53,54,55,56,57,58,59,60,61,62]. Atrial fibrillation occurs in approximately 6–16% patients, predominantly in the first 6 months of treatment and is mostly grade 1–2 [63,64]. Discontinuation of ibrutinib therapy does not affect the resolution of the event; however, if worsening to grade 3 or higher occurs, ibrutinib should be temporarily withheld until the adverse rhythm is resolved [65]. It should be noted that potentially lethal ventricular arrhythmias were also noted during ibrutinib treatment and could contribute to sudden unexpected deaths [66]. The exact mechanism of atrial fibrillation under ibrutinib therapy remains unknown; however, the off-target inhibition of the cardiac PI3K isoform and alteration of late cardiac sodium currents has been proposed [67,68].

The increased risk of bleeding is linked with acquired platelet dysfunction and is manifested particularly as skin–mucosal hemorrhagic diathesis (predominantly petechiae and ecchymoses) affecting approximately 20–40% of patients [16,51,56]. The exact mechanism of this particular toxicity remains unclear, however BTK was shown to be involved in platelet glycoprotines GP1b (via the von Willebrand factor) and GPIV (via collagen) signaling [69,70]. Major bleeding events were estimated to occur in 1–4% of patients, however a recently performed analysis of 1768 patients treated with ibrutinib revealed a similar risk of major bleeding for non-ibrutinib treated patients when data were adjusted for exposure time to the drug (3.2 vs. 3.1 per 1000 person-months) [71].

Diarrhea based on clinical reports is the most often noted mild adverse event during ibrutinib treatment. Its pathogenesis may be multifactorial, but taking into account that it is also frequently observed in patients under EGFR inhibitors, the off-target inhibition of this kinase by ibrutinib may be a contributing factor in the etiology of this adverse event [72]. 

Despite, the well characterized toxicity profile of the drug, analysis of clinical trials and real-world data revealed that in particular patient cohorts, up to approximately 30% patients discontinue ibrutinib therapy due to toxicity [57,61,62,73,74,75,76].

## 6. Resistance Mechanisms and Clinical Implications

Despite the clinical efficacy of ibrutinib, both primary and acquired resistance to the agent has been described in clinical trials and real-world patient settings [13,16,57,58,74,77,78,79,80]. Primary resistance to ibrutinib, accounting for approximately 13–30% of cases, occurs only in patients with CLL with no initial response and mostly is observed in RR-CLL cases with an underlying possible Richter transformation. These observations are in line with results showing that patients with germinal center B cell (GCB)-DLBCL almost uniformly lack responsiveness to ibrutinib [77,81,82]. In the case of DLBCL, overexpression of the CD79B was also reported to be responsible for primary ibrutinib resistance; however, such a mechanism was not reported in CLL [83]. Secondary resistance to BTK inhibitors is far better characterized in CLL, where it can be manifested as a Richter transformation during the first year of therapy or as progressive CLL [62,78,84,85,86]. It was shown that CLL cells under ibrutinib pressure are prone to clonal shifts as determined by whole-exome sequencing studies [87,88]. So far, complex karyotype, 17p deletion and BCL6 abnormalities were shown to be risk factors for acquired secondary resistance [84,89]. Whole-exome sequencing analysis of patients with late relapses showed acquired mutations in BTK at the binding site of ibrutinib (C481) with a cysteine to serine mutation, and PLCγ2, the kinase immediately downstream of BTK, where multiple activating mutations were identified [87,88,90].

The functional characterization of these mutations demonstrated that BTK C481S mutation reduces the binding affinity of ibrutinib for BTK, leading to reversible inhibition. Because of the relatively short half-life of ibrutinib, it has been confirmed that patients who relapse with the C481S mutation show expression of phosphorylated BTK which is not inhibited by the administration of ibrutinib [91]. The serine residues in the C481 position prevents the ibrutinib covalent from binding, making the bond reversible and reducing the ability to inhibit the mutant form of BTK, and therefore reducing its clinical activity [91]. Importantly, these mutations may be found in CLL cells up to one year before the clinical relapse occurs, allowing potentially for preemptive therapy modification [88,92,93]. Additional mutations at the T474 (T474I and T474S) locus leading to diminished ibrutinib selectivity and affinity have been described [94]. Recently, a substitution mutation T316A coding the non-kinase SH2 domain was identified. It is predicted that this results in diminished BTK interactions with BLNK and other proteins that drive ibrutinib resistance [95]. Nevertheless, mutations in BTK may occur simultaneously with *PLCG2*, which may pose problems when selecting proper next line treatment [88,93,96].

Mutations identified in *PLCG2* have all been demonstrated to potentially gain a function, allowing BCR signaling activation in the presence of inactive BTK [88,93,97]. In *PLCG2*, the R665W mutation causes missense alteration leading to BTK-independent activation after BCR engagement, allowing it to bypass the BTK. Furthermore, BCR proximal kinases SYK and LYN are critical for the activation of mutant PLCγ2, indicating that the SYK and LYN blockades may potentially have clinical relevance in overcoming mutated PLCγ2 acquired ibrutinib resistance [97]. 

In addition to the point mutations in the above-mentioned genes, deletion of the short arm of chromosome 8 has also been connected with secondary resistance to ibrutinib [87]. In the series of five patients, Burger et al. reported expansion of clones harboring del(8p) with additional driver mutations (*EP300*, *MLL2* and *EIF2A*), and interestingly, this clonal shift resulted in one patient with trans-differentiation into CD19-negative histiocytic sarcoma [87]. Del(8p) resulted in haploinsufficiency of the TNF-related apoptosis-inducing ligand receptor (TRAIL-R), leading to TRAIL insensitivity. Combined administration of TRAIL and ibrutinib resulted in CLL cells’ diminished viability in the non-del(8p) samples; however, in the del(8p), this was not observed [87]. The association of del(8p) with point mutations in *RPS15* and *SF3B1* could potentially be linked with an additional clonal advantage of these cells upon ibrutinib treatment [87]. Recently, an enrichment in *SF3B1*, *MGA*, *BIRC3*, *NFKBIE, CARD11* and *XPO1* point mutations was noted in CLL samples with an acquired ibrutinib resistance [88,90].

Cosson et al. described a gain of the short arm of chromosome 2 (2p) leading to exportin-1 gene (*XPO1*) overexpression [98]. *XPO1* regulates the transport of several cycle regulatory proteins, e.g., p53, FOXO and retinoblastoma (pRb) from the nucleus to the cytoplasm [99,100]. Overexpression of *XPO1* leads to an efflux of the above-mentioned proteins from the cell nucleus, preventing their cell regulatory capabilities, and was linked with resistance to fludarabine–cyclophosphamide–rituximab (FCR) immunochemotherapy and ibrutinib [88,98].

Recently, Spina et al., based on the observations of 31 high-risk CLL patients treated with ibrutinib within the IOSI-EMA-001 study (#NCT0287617), proposed that clonal shifts promote cells with constant activations of AKT and ERK and non-canonical NF-κB pathways, preventing their death [101]. ERK activation was already shown to mediate ibrutinib resistance in MW [102].

## 7. Alternative Irreversible BTK Inhibitors

Ibrutinib has enormously changed the outcome of CLL patients. Nevertheless, acquired ibrutinib resistance and disease progression remain the real challenge in CLL treatment. Moreover, off-target kinase activity may contribute to adverse effects, which are the most common reason for discontinuation in clinical practice. Therefore, more specific BTKi could possibly be better tolerated while maintaining high clinical efficacy. The most important compounds with potential activity as the post-ibrutinib therapy are listed in Table 1.

### 7.1. Acalabrutinib

Acalabrutinib (ACP-196) is an oral, highly selective, irreversible and covalent BTK inhibitor which has been proven to inhibit off-target kinases, such as EGFR and ITK, to a lesser extent than ibrutinib [103,104]. In the phase I/II ACE-CL-001 trial (#NCT02029443), acalabrutinib was administered to 134 RR-CLL patients after at least one prior treatment regimen [105]. The ORR reached 85%, and inclusion of partial responses with lymphocytisis (PR-L) increased it to 93%, while the median PFS was not reached. The toxicity profile was better than that noted for ibrutinib with a lower frequency of typical ibrutinib adverse events (AEs)—hypertension of any grade occurred in 11% (grade ≥3 in 2%), while atrial fibrillation in 3% (grade ≥3 in 2%). No grade ≥3 bleeding episodes occurred. 

In an ongoing phase 2 study, acalabrutinib monotherapy has been indicated in patients with RR-CLL (n = 30) and high-risk TN-CLL (n = 16) patients [106]. Forty-six patients were enrolled, of which 39% had bulky lymph nodes ≥5 cm, 76% had unmutated IGHV, 21% had del(17p) and 21% had *TP53* mutation. The median time of observation was 33 months. The ORR was 90%, and after 20 months of follow-up, 89% of patients (41/46) remained on treatment [106]. 

Taking into consideration the lower off-target toxicity of acalabrutinib in comparison to ibrutinib, acalabrutinib has been studied for ibrutinib-intolerant patients. In the subanalysis of the ACE-CL-001 trial, 33 patients were in the ibrutinib intolerant cohort [107]. Acalabrutinib was discontinued in two patients (6%) due to AE (one unrelated G5 fungal infection and one unrelated G3 metastatic endometrial cancer). Several studies on setting acalabrutinib in monotherapy or in combination with other agents as an alternative to ibrutinib intolerance have been ongoing, including those which directly compare these two agents (phase III trial #NCT02477696). In summary, results to date demonstrate that acalabrutinib is highly active as both salvage therapy for CLL and for patients with ibrutinib intolerance. Acalabrutinib seems to display better tolerability than ibrutinib; however, the results of a head to head comparison of both agents are eagerly awaited.

### 7.2. Zanubrutinib

Zanubrutinib (BGB-3111) is another irreversible second generation BTKi which is also more selective for BTK than ibrutinib [108]. A phase I study by Tam et al. established a dose of 320 mg daily (either QD or 160 mg BID) as inhibitory to 99.5% of nodal BTK [109]. After a median follow-up of 13.7 months, 89 (94.7%) CLL/SLL patients have remained on treatment. Grade 3 neutropenia occurred in two patients, whereas one patient suffered from subcutaneous hemorrhage. Seventy-eight patients were evaluable for the efficacy analysis. The ORR was 96.2% and the estimate 12 month PFS rate was 100%. A head to head comparison between zanubrutinib and ibrutinib is conducted in an ongoing trial for patients with relapsed and refractory CLL or SLL (#NCT03734016). 

### 7.3. Tirabrutinib

Tirabrutinib (ONO/GS-4059) belongs to second-generation, irreversible BTKi. Similarly to acalabrutinib, the drug displays a higher degree of selectivity to BTK than to other TEC kinases [110,111]. In a multicenter phase I dose-escalation study (#NCT01659255), tirabrutinib was indicated in 90 patients with relapsed and refractory B cell malignancies, including 28 patients with CLL [111]. Eleven patients (39%) were refractory to chemotherapy and nine (36%) had del(17p), whereas 13 (52%) had *TP53* mutation. There were nine dose-escalation cohorts ranging from 20 mg to 600 mg once daily. The median follow-up for CLL was 560 days. Twenty-four of 25 evaluable CLL patients (96%) responded to tirabrutinib regardless of tested dose levels, and 24 (96%) patients had an objective lymph node response. Similarly, all of the patients with del(17p) or *TP53* mutation responded to the treatment. At the data cut-off, 75% continued with tirabrutinib. Seven patients discontinued treatment—five due to AE and two due to progressive disease. The majority of Grade 3/4 AEs were mainly hematologic (anemia in 11% of patients and thrombocytopenia in 7%) and recovered spontaneously during therapy. One patient experienced a grade 3 bleeding event (spontaneous muscle hematoma). No arrhythmias were observed [111]. The efficacy and toxicity profile from the early-phase trial seems to be more favorable than that of ibrutinib.

## 8. Reversible BTK Inhibitors

Acalabrutinib, zanubrutinib and tirabrutinib display higher BTK selectivity, however they all bind covalently via C481 residue. Reversible BTK inhibition is a promising strategy to combat progressive CLL, and multikinase inhibition demonstrates superior efficacy to targeted ibrutinib therapy in the setting of Richter transformations. In summary, given the activity, regardless of C481S mutation, this class of agents may be an alternative for ibrutinib resistance.

### 8.1. GDC-0853

GDC-0853 is an oral, highly selective and reversible BTK inhibitor which binds independently from C481 [112,113]. GDC-0853 suppresses downstream BCR signaling, resulting in downregulation of the NF-κB pathway and inhibition of cell proliferation [112,113]. In a phase I study, GDC-0853 was indicated in 24 patients with relapsed or refractory non-Hodgkin lymphoma (10 patients) or CLL (14 patients) [114]. Six patients were positive for C481S mutation. There were three cohorts taking 100, 200, or 400 mg once daily. One third of patients (eight out of 24) responded: one had CR and seven had PR or PR-L. Common AEs included fatigue (37%), nausea (33%), diarrhea (29%), thrombocytopenia (25%), headache (20%), abdominal pain, cough and dizziness (16%, each). Nine serious AEs were reported in five patients, of whom two had fatal outcomes (confirmed H1N1 influenza and influenza pneumonia). One patient with a C481S mutation achieved PR and another two patients had a decrease in size of target tumors (–23% and –44%). These data demonstrate that GDC-0853 was generally well-tolerated and had the potency to overcome C481S-mediated resistance.

### 8.2. Vecabrutinib (SNS-062)

Vecabrutinib is a novel reversible BTK and ITK inhibitor which has the ability to inhibit BTK, even in the presence of C481S mutations. Moreover, it seems to inhibit EGFR to a lesser extent than ibrutinib [115,116]. Recently, the preliminary results of the ongoing phase Ib/II study of vecabrutinib have been published (#NCT030376450) [117]. Nine patients (6 CLL, 2 MCL, 1 WM) have been treated in the phase Ib portion of the study (25 mg BID, n = 3; 50 mg BID, n = 6). The cohort was heavily pretreated with median of five prior regimens. All patients had history of BTKi treatment (eight with ibrutinib, one with acalabrutinib), four patients received venetoclax and two had chimeric antigen receptor T-cells (CAR-T). At the baseline, 67% (six out of nine) of patients had del(17p) or *TP53* mutations. The most common grade ≥3 AEs included anemia, neutropenia and increased alanine transaminase (ALT) activity and occurred in one patient. The results indicate that vecabrutinib is rather well tolerated; however, final results with a response assessment are needed to verify its efficacy.

### 8.3. LOXO-305

LOXO-305 is a recently described, non-covalent, reversible and selective BTKi [118]. In preclinical models, LOXO-305 has displayed potent BTK inhibition regardless of the presence of C481S mutation. Moreover, it has minimal off-target kinase efficacy and non-kinase activity. The clinical trials are at very early stages, a phase I/II study of oral LOXO-305 in patients with RR-CLL/SLL and non-Hodgkin lymphoma (NHL) (#NCT03740529) has recently started.

### 8.4. ARQ-531

ARQ-531 binds reversibly and in a non-covalent way to BTK in the ATP binding region, omitting C481 [119]. Additionally, ARQ-531 inhibits other kinases in the BCR signaling pathway, e.g., LYN kinase from the SRC family and MEK1 kinase which, in turn, blocks the downstream ERK signaling. Because of the multiple target sites in animal models and patients’ samples, ARQ-531 was able to inhibit BCR signaling in cells harboring either C481S or *PLCG2* mutations. A phase I clinical trial is ongoing to assess the safety and toxicity, and to establish dosing (#NCT 03162536). The preliminary results from the 16 patients with relapsed and refractory B cell malignancies with a history of BTKi treatment, including 12 patients with CLL, were published recently by Woyach et al. [120]. Patients were treated with doses up to 30 mg orally. The most common reported AEs were diarrhea, nausea, vomiting, fatigue, neutropenia and thrombocytopenia, hypernatremia, facial paralysis and headache, all reported in one patient (6.3%). Grade 3 AEs occurred in two patients (one lipase level elevation and thrombocytopenia). Twelve patients received at least one dose of the study drug and were available for response assessment, five achieved stable disease (SD) and seven had a progressive disease (PD). The results seem to show a manageable toxicity profile; however, the follow-up for objective responses is awaited. 

## 9. Alternate Kinases Inhibitors and Drugs Potentially Overcoming Ibrutinib Resistance

### 9.1. Venetoclax

Venetoclax (ABT-199) is a BCL2 antagonist capable of inducing CLL rapid cell apoptosis independently of microenvironmental signals and signaling pathways [4,121,122]. In the pivotal trial (#NCT01328626), monotherapy with the agent showed remarkable activity in heavily pretreated RR-CLL, achieving a 77% ORR, including a 23% CR [122]. The addition of rituximab even potentiated its clinical efficacy, and in a cohort of 49 RR-CLL patients led to an 88% ORR with a 31% CR (#NCT01682616) [123]. Most importantly, venetoclax is also useful in the clinical setting of ibrutinib resistance based on data derived from clinical trials and real-world experience [60,124,125,126,127]. In a group, 127 RR-CLL patients, of whom 91 were treated with ibrutinib as the last BCR inhibitor, were treated with venetoclax in monotherapy. In the analyzed group, 59/91 (64.8%) patients achieved response to venetoclax, and within the median follow-up of 14 months, 17 (19%) patients died; however, only seven deaths were attributable to disease progression (#NCT02141282) [126]. These results are supported by the results obtained in US, Italian and UK real-world cohorts [60,124,127]. In the US cohort of 683 patients treated with ibrutinib or idelalisib initially, upon kinase inhibitor failure, either chemoimmunotherapy, alternate inhibitor or venetoclax was administered. Based on PFS data, a switch to an alternate kinase inhibitor or venetoclax was most beneficial when compared to immunochemotherapy. Furthermore, patients who discontinued ibrutinib due to progression or toxicity had marginally improved outcomes if they received venetoclax (79% ORR) compared to idelalisib (46% ORR) [60]. The efficacy of venetoclax was also confirmed in the 105 RR-CLL patients in the UK cohort. In this group 60% of patients received the BTK inhibitor solely and 10% were treated with both the BTK and PI3K inhibitor. Treatment was stopped due to disease progression in 54% and 44% patients, respectively. In the BTK inhibitor cohort, the 85% ORR (23% CR) was achieved. Moreover, venetoclax was also active in patients exposed to both BTK and PI3K inhibitors, in which an 80% ORR (23% CR) was reached [127]. In the Italian cohort, of the 76 evaluable patients, 52 had received venetoclax after one BCR inhibitor (37 ibrutinib; 15 idelalisib n = 15), and 24 received two BCR inhibitors. Although the ORR following treatment with single BCR inhibitor was reasonably high in the analyzed cohort, reaching 74%, once the etiology of the BCR discontinuation was analyzed, the results strongly differed. The highest response to venetoclax (91% ORR) was noted in patients who discontinued BCR due to adverse events; however, in cases where the therapy was stopped due to disease progression, only a 49% ORR was observed [124]. Venetoclax activity diminished, however, if administered following two or more BCR inhibitors. In the cohort of 28 such patients, the ORR reached only 43%; however, the difference between cases ceasing therapy due to adverse events and disease progression was much lower than in the Italian cohort (50% vs. 38%) [128]. Considering the above-mentioned data, venetoclax is so far the best treatment modality for CLL cases following failure of BCR inhibitor therapy, and should be administered as a first treatment choice due to the observed diminished activity with increasing lines of selective agents. Whether preemptive addition of venetoclax to ibrutinib therapy in the case of increased ibrutinib refractory clones (e.g., harboring BTK mutations) could prevent clinical disease progression is still an open question. Nevertheless, the combination of ibrutinib and venetoclax was shown to be well tolerated and effective in cohorts of TN-CLL and RR-CLL cases [129,130]. In the CLARITY study, a response was noted in 47 out of 53 RR-CLL patients treated with such a combination (89%), with 51% achieving CR and 36% bone marrow minimal residual disease (MRD) negativity [130].

### 9.2. SYK Inhibitors

Spleen tyrosine kinase (SYK) is one of the downstream kinases in BCR signaling. It mediates signaling, proliferation, migration and survival of CLL cells [131]. Entospletinib (GS-9973) is an oral selective inhibitor of SYK that was designed for overcoming BTKi resistance [131]. In an open-label phase 2 trial (#NCT01799889), including patients with RR-CLL (n = 41) or NHL (n = 145) with no history of BTKi treatment, 400 mg of entospletinib was taken orally twice daily [132]. The ORR was 61.0% with a 6 month PFS rate of 70.1%, while the median PFS was 13.8 months. Severe AEs occurred in 29% of patients, with dyspnea (4.2%), pneumonia (4.2%), febrile neutropenia (3.2%), dehydration (2.7%) and pyrexia (2.2%) as the most common. Grade 3 or 4 AEs were neutropenia (14.5%) and anemia (8,1%). Non-hematological AEs included fatigue (10.2%), increased alanine aminotransferase (12.9%) and aspartate aminotransferase (10.2%) activity [132].

Another phase 2 trial (#NCT01799889) was designed for patients with RR-CLL, including RT, who were previously treated with a BTKi [133]. Forty-nine patients were enrolled, including eight with RT. Ibrutinib was the most often used BTKi in their disease course (37 patients; 75.5%). Entospletinib was administered in a dose of 400 mg BID. Sixteen patients (32.7%) achieved PR and 21 (42.9%) had SD; however, no complete responses were noted. The median PFS was 5.6 months. AEs occurred in all patients and were similar to the previous study [133]. Entospletinib in monotherapy demonstrates clinical activity for patients with RR-CLL, including those who have relapsed after BTKi therapy.

In addition, Cheng et al. showed that C481S mutation-mediated resistance may be abolished with other SYK inhibitors [91]. In the in vitro model, CLL cells were treated with two SYK inhibitors: cerdulatinib (a dual inhibitor of SYK/JAK (PRT062070) and the highly specific SYK inhibitor PRT060318) and dasatinib (a LYN and BTK inhibitor). Both of the inhibitors with activity against SYK (PRT062070 and PRT060318), as well as dasatinib, led to a complete block of CLL cells’ proliferation [91].

Cerdulatinib is a dual SYK/JAK kinase inhibitor which was shown to have antitumor activity in comparison to either target alone in preclinical studies [91,134]. The phase I dose-escalation study of cerdulatinib in 43 patients with relapsed and refractory B-cell malignancies detailed the pharmacokinetics, safety and initial efficacy [135]. Of the dosed patients, eight had CLL/SLL, 13 had follicular lymphoma and 22 had aggressive B cell NHL. The most common grade 3 or 4 AEs were anemia (16%), fatigue (14%) and diarrhea (9%). Cerdulatinib was temporarily interrupted due to AE in 13 patients. The most common of these were fatigue (n = 5) and gastrointestinal events (n = 3). Five patients achieved a partial response (CLL, n = 3; FL, n = 1; transformed FL grade 3B, n = 1). All patients with CLL who achieved a response were on therapy for more than 200 days. Five patients did not respond to the treatment: two because of Pneumocystis pneumonia, one patient had grade 5 pneumonia and another two had rapidly progressed [135]. The phase II study is currently ongoing to assess the efficacy of cerdulatinib (#NCT01994382).

### 9.3. PI3K Inhibitors

Duvelisib (IPI-145) is an oral dual inhibitor of PI3Kδ and PI3Kγ [136]. Duvelisib inhibits the PI3Kδ isoform, which affects cell proliferation and survival, and the PI3Kγ isoform, which plays a role in the interactions with the microenvironment. The phase I trial (#NCT01476657) included 210 patients with various hematological malignancies: most patients had RR-CLL (n = 55), then T-cell lymphoma (n = 35), indolent NHL (n = 31), aggressive NHL (n = 26) and other diagnoses (n = 71) [137]. Based on MTDs, the dose of duvelisib in the expansion phase was 25 to 75 mg orally BID. Clinical responses were observed in every dose and disease group. The ORR in RR-CLL was 56%, including one CR. The median time to response was approximately 1.8 months. Grade 3 or 4 AEs occurred in 84% of patients. Among hematological AEs, neutropenia was the most frequent (in 39% of patients in any grade and in 32% of patients in grade ≥3). Anemia and thrombocytopenia occurred in 14% of patients from each grade. Non-hematological AEs included increased alanine transaminase (20%), aspartate transaminase activity (15%), diarrhea (11%) and pneumonia (10%) [137]. The results from the phase II trial (#*NCT01882803*) of duvelisib were recently published by Flinn et al. [138]. The trial enrolled 129 patients with indolent, refractory NHL FL (83 patients), SLL (28 patients) and MZL (18 patients). Twenty-five milligrams of duvelisib was administered orally BID in 28 day cycles until progression, unacceptable toxicity or death, and the ORR was 47.3%, reaching 67.9% in the SLL group. The most frequent AEs were diarrhea (48.8%), nausea (29.5%), neutropenia (28.7%), fatigue (27.9%) and cough (27.1%). Grade 3 or 4 AEs occurred in 88.4% of patients, and included neutropenia (24.8%), diarrhea (14.7%), anemia (14.7%) and thrombocytopenia (11.6%) [138]. The studies showed the potential efficacy and manageable safety profile of duvelisib in RR-NHLs. However, neither idelalisib nor duvelisib can override the signal transduction of PLCγ2 with the R665W mutation [97]. 

### 9.4. Compound 1

A novel class of BTK targeting agents has been recently designed which inhibits both BTK and PI3Kδ [139]. It has been hypothesized that the inhibition of these pathways simultaneously could result in deeper responses or overcoming resistance in comparison to using a single agent. Compound 1 is the first in the class of drugs which have been undergoing biological tests [139].

### 9.5. Exportin-1 Inhibitors

Selinexor (KPT330) is an inhibitor of a nuclear exporter of tumor suppressing proteins, e.g., p53 [140]. Selinexor inhibits activation of downstream BCR kinases and suppresses BTK gene expression [141]. In a phase I trial, 79 patients with various NHLs were enrolled (#NCT01607892) [142]. The most common types of NHLs were DLBCL (n = 43), RT (n = 8), FL (n = 9) and CLL (n = 7). In the dose-expansion phase, selinexor was administered at 35 or 60 mg/m^2^. The most frequent AEs in grade ≥3 were thrombocytopenia (47%), neutropenia (32%), anemia (27%), leukopenia (16%), fatigue (11%) and hyponatremia (10%). Objective responses included four CR and 18 PR across the various NHL subtypes. A dose of 35 mg/m^2^ (60 mg) was identified as the recommended phase 2 dose (RP2D) [142]. The study is, as of now, the only clinical trial that enrolled CLL patients. A phase II trial has been ongoing in DLBCL. Another interesting option in pre-clinical trials is attempting to combine selinexor with ibrutinib. Hing et al. revealed that selinexor and ibrutinib act synergistically in mouse CLL models and in samples from patients [143].

### 9.6. Mammalian Target of Rapamycin (mTOR) Pathway Inhibition

mTOR is a downstream kinase in the PI3K-AKT pathway which promotes CLL cell proliferation. A novel class of drugs and a novel inhibitor of the mammalian target of rapamycin kinase (TORK) and DNA-dependent protein kinase (DNA-PK), CC-115 was evaluated in vitro and led to caspase-dependent cell killing irrespective of the p53, ATM, NOTCH1 or SF3B1 status [144]. This was confirmed in CLL samples obtained from patients with an acquired resistance to ibrutinib treatment [144]. The phase Ia/Ib trial (#NCT01353625) is already ongoing, but its preliminary results have revealed the efficacy of CC-115 in eight patients with RR-CLL with the ATM mutation. Seven patients achieved a decrease in lymphadenopathy, one achieved PR and three achieved PR-L [144]. 

### 9.7. Hsp90 Inhibition

Hsp90 is a molecular chaperone that stabilizes client proteins and prevents their proteasomal degradation [145]. SNX-2112 and its prodrug SNX-5422 are novel, highly selective Hsp90 inhibitors that have been revealed to be active in vitro in multiple myeloma and other hematological malignant cell lines [146]. This resulted in the translation of SNX-5422 in a phase I clinical trial (#NCT02973399) to assess whether the combination of SNX-5422 with ibrutinib will provide a better clinical response in subjects who have residual disease than ibrutinib therapy, but not PD. The results of the trial have not been published yet. Another Hsp90 inhibitor, AUY922, was found to initiate the degradation of BTK and IκB kinases in ibrutinib-resistant MCL cell lines [147].

### 9.8. Other Potential Molecules for BTKi-Resistant Patients

The bromodomain and extraterminal family of bromodomains (BET) have been targeted by many antagonists in various types of tumors [148]. The BET family regulates gene transcription [149]. JQ1 is an inhibitor of one of the BET family, BRD4. In preclinical studies, it inhibits the transcription of BCL2, c-MYC and as such mediates the apoptosis of acute myelogenous leukemia cells [150]. In MCL cell lines, it was shown to decrease BTK, PLCγ2 and AKT levels, and to induce apoptosis of the cells [151]. Moreover, the combination of JQ1 and ibrutinib worked synergistically in inducing apoptosis [151].

Pimasertib (AS703026/MSC1936369B) is a selective MEK1/2 inhibitor which has been extensively tested in clinical trials in solid tumors. Recently, in vitro research revealed its synergism with a combination of idelalisib or ibrutinib in DLBCL and MCL cell lines [152].

Afuresertib (GSK2110183) is an oral inhibitor targeting AKT which was shown to inhibit cell proliferation in several hematologic malignancies’ cell lines [153]. In a phase I–II study, afusertib demonstrated activity in heavily pretreated multiple myeloma and NHL patients [153].

The activity of nutlin-3, an MDM inhibitor, in combination with ibrutinib was evaluated in vitro in a panel of B cell leukemia cell lines and in CLL patient samples [154]. The study revealed enhanced apoptosis, even in the samples carrying del(17p) and/or TP53 mutations [154].

## 10. Chimeric Antigen Receptor T Cells

T cells with the modified chimeric antigen receptor (CAR) were shown to overcome immunological tolerance and mediate tumor rejection in patients with acute lymphoblastic leukemia (ALL) and DLBCL [155]. Although the use of CAR T cells in CLL is limited due to widespread of new effective treatment modalities, the safety and efficacy data suggest that they may overcome BCR receptor inhibitor resistance [156,157,158]. Initial studies by Porter et al. on heavily pretreated RR-CLL patients have generated encouraging preliminary data. The ORR was eight of 14 (57%), with four CR and four PR. The in vivo expansion of the CAR T cells correlated with clinical responses, and the CAR T cells persisted and remained functional beyond 4 years in the first two patients achieving CR. At the time of publication, no patient in CR has relapsed [159]. In a group of 24 RR-CLL patients, anti-CD19 CAR-T cells showed promising clinical activity upon ibrutinib resistance (NCT01865617) [157]. In the analyzed group, 19 progressed under ibrutinib, three had ibrutinib intolerance and two did not experience progression while receiving ibrutinib. Moreover, six patients were venetoclax refractory and 23 had a complex karyotype and/or 17p deletion. In the analyzed patient cohort, after four weeks following CAR-T cells’ infusion, a 71% ORR was achieved. Twenty of 24 patients received cyclophosphamide and fludarabine lymphodepletion before CAR-T cells’ infusion. Among this group, restaging was performed in 19 patients (including 16 ibrutinib refractory patients) which showed a 74% ORR (21% CR and 53% PR). Minimal residual disease (MRD) negativity based on *IGH* sequencing was shown in seven out of 12 (58%) assessed patients. In these patients, a 100% PFS and OS rate under 6.6 months’ follow-up was noted. In the whole cohort, the median PFS was 8.6 months, whereas the median OS was not reached [157]. In the TRANSCEND CLL 004 phase I/II trial, anti-CD19 CAR-T cells were administered to 10 patients with RR-CLL with 70% classified as having high risk disease. The median of prior therapies in this cohort was four (range 3–8), while 90% of patients had received ibrutinib and 60% patients had been treated with both venetoclax and ibrutinib [158]. At 30 days post-dose, in six of eight (75%) patients an ORR was achieved, including four (50%) CR. Furthermore, six of seven (85.7%) patients were characterized by MRD negativity. In five patients eligible for response assessment following 3 months, four had a sustained response with MRD negativity, however one Richter transformation was noted [158]. Considering the above-mentioned results, despite a short observation time and a low number of treated patients, the use of CAR-T cells following ibrutinib failure poses an interesting treatment modality.

## 11. Conclusions and Future Directions

The development of ibrutinib and other selective BTK inhibitors increased the therapeutic armamentarium for CLL. However, despite high clinical activity of ibrutinib, patients discontinue the treatment due to loss of initial response and intolerance. Considering the increasing widespread use of ibrutinib in RR-CLL, and its potential shift to a frontline setting, the number of patients with BTK inhibitor resistance or failure will likely increase. Hopefully, potential strategies for overcoming this clinical problem are possible. First, the use of venetoclax alone or in drug combinations is so far the most promising option following ibrutinib therapy failure. Secondly, a combination of ibrutinib with chemoimmunotherapy for a restricted period of time could potentially limit the clonal disease evolution, minimizing the development of ibrutinib-resistant clones. This approach, however, is currently under evaluation, and further prolonged observations are necessary. Thirdly, use of CAR-T cells present a viable therapy of disease progression upon resistance to ibrutinib, regardless of the presence of a Richter transformation; however, the data are scarce and not as promising as DLBCL and ALL. An increasing number of translational data may help to guide post-ibrutinib therapies depending on the disease’s clinical presentation and molecular data.

## Figures and Tables

**Figure 1 cancers-11-01834-f001:**
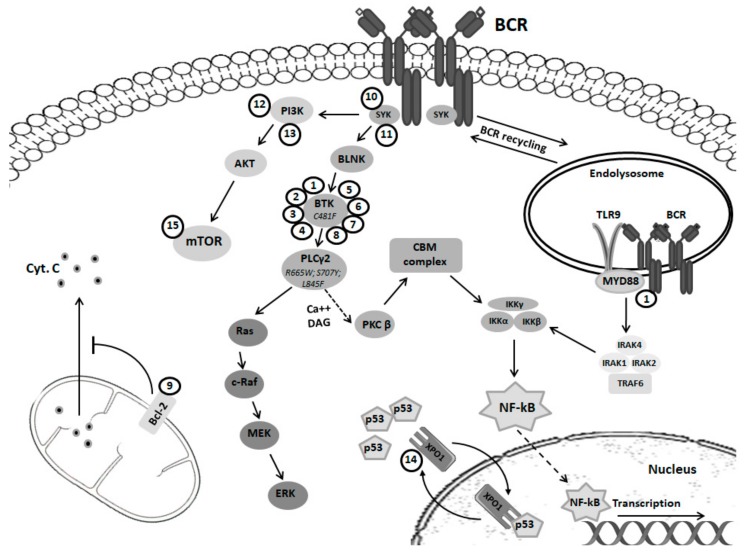
Overview of the B-cell receptor pathway. Shown are the B cell receptor (BCR) and signaling intermediates engaged in signal propagation following binding of the BCR to antigen. Numbers indicate inhibitors targeting particular components of the BCR signaling: (1) ibrutinib; (2) acalabrutinib; (3) zanabrutinib; (4) tirabrutinib; (5) GDC-0853; (6) vecabrutinib; (7) LOXO-305; (8) ARQ-531; (9) venetoclax; (10) entospletinib; (11) cerdulatinib; (12) duvelisib; (13) idelalisib; (14) selinexor; (15) CC-115. See main text for details. Bcl-2, B-cell lymphoma 2 protein; BLNK, B-cell linker protein; BTK, Bruton tyrosine kinase; CBM, CARD11–BCL-10–MALT1; Cyt.C, cytochrome C; DAG, diacylglycerol; ERK, extracellular signal–regulated kinase; IKK, inhibitor of NF-κB kinase; IRAKs, interleukin-1 receptor-associated kinases; MEK, mitogen-activated protein kinase; mTOR, mammalian target of rapamycin; MYD88, myeloid differentiation primary response 88; NF-κB, nuclear factor-κB; PI3K, phosphoinositide 3-kinase; PKCβ, protein kinase Cβ; PLCγ, phospholipase Cγ; SYK, spleen tyrosine kinase; TLR9, toll-like receptor 9; TRAF6, TNF receptor-associated factor 6; XPO1, exportin 1.

**Table 1 cancers-11-01834-t001:** Drugs with the potency to overcome ibrutinib resistance/intolerance.

Drug (Symbol)	Target	Mechanism	Advantage Over Ibrutinib	Phase and Clinical Trial Number	Head to Head Comparison to Ibrutinib
Acalabrutinib (ACP-196)	BTK	Irreversible and covalent inhibitor	Minimal off-target inhibitory activity, no anti-platelet activity, effective to ibrutinib intolerance	III NCT04008706	NCT02477696
Zanubrutinib (BGB-3111)	BTK	Irreversible and covalent inhibitor	Less inhibitory to ITK	III NCT04116437 NCT03336333	NCT03734016
Tirabrutinib (ONO/GS-4059)	BTK	Irreversible, covalent inhibitor	Less inhibitory to TEC	III NCT02457559	No
GDC-0853	BTK	Reversible, non-covalent inhibitor	Overcomes C481S mutation, effective to ibrutinib resistance	I NCT01991184	No
Vecabrutinib (SNS-062)	BTK	Reversible, non-covalent inhibitor	Overcomes C481S mutation, effective to ibrutinib resistance, less inhibitory to EGFR	I/II NCT03037645	No
LOXO-305	BTK	Reversible, non-covalent	Overcomes C481S mutation	I/II NCT03740529	No
ARQ-531	BTK LYN MEK	Reversible, non-covalent	Overcomes C481S and *PLCG2* mutations	I/II NCT03740529	No
Entospletinib (GS-9973)	SYK	ATP-competitive, selective inhibitor	Overcomes *PLCG2* mutations	II NCT01796470 NCT01799889	No
Cerdulatinib (PRT062070)	SYK/JAK	ATP-competitive inhibitor	Overcomes C481S mutation and inhibits microenvironment support for CLL cells	I/II NCT01994382	No
Duvelisib (IPI-145)	PI3Kγ PI3Kδ	ATP-binding region	Inhibits microenvironment support for CLL cells	III NCT02004522	No
Selinexor (KPT330)	XPO-1	Reversible, covalent inhibitor	Overcomes C481S mutation, suppresses *BTK* gene, inhibits activation of downstream BCR targets ERK and AKT	II NCT02227251	No
CC-115	mTOR	Inhibits auto-phosphorylation of the catalytic site	Overcomes ibrutinib resistance	I NCT01353625	No

AKT, protein kinase B; ATP, adenosine triphosphate; BMX, cytoplasmic tyrosine–protein kinase BMX; BTK, Bruton kinase; CLL, chronic lymphocytic leukemia; EGFR, epidermal growth factor receptor; ERK, extracellular signal-regulated kinase; ITK, interleukin 2–inducible T cell kinase; JAK, Janus kinase; LYN, thyrosine-protein kinase; mTOR, mammalian target of rapamycin; PI3K, phosphoinositide 3-kinase; SYK, spleen tyrosine kinase; TEC, Tec protein tyrosine kinase; XPO-1, Exportin-1.

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
