# Peer review of "Overcoming Ibrutinib Resistance in Chronic Lymphocytic Leukemia"

_cancers, 2019, doi:10.3390/cancers11121834_

Round 1
Reviewer 1 Report
In this manuscript, chronic lymphocytic leukemia inhibitor drugs are reviewed and discussed for exploring the solution of resistance. There are multiple drugs targeting Bruton tyrosine kinase for inhibition, and there are alternative drugs when this class doesn’t work as well. It is a favorable idea to compare them for the sake of better treatment for CLL patients. In this study, the molecular mechanism of B cell receptor signaling pathways is reviewed, and drugs targeting these pathways are well sorted in Figure 1, which is informative and impressive. One of the front line drugs for inhibiting BTK, ibrutinib, is heavily discussed, including the working mechanism, clinical activity, adverse effects, and resistance mechanisms. Other BTK inhibitors are also discussed. Alternate kinases inhibitors and drugs potentially overcoming ibrutinib resistance are well discussed.
Over all it is a well organized and thoroughly discussed study. I have only one suggestion which may improve its integrity for publication. Off-target activity of ibrutinib was discussed in Part 3, which included several tyrosine kinases. How are these pathways connected to the adverse events discussed in Part 5? It will help to understand the molecular basis of it side effects.
Author Response
Thank you for this suggestion. So far only some of ibrutinib potential mechanisms leading to adverse event were identified. In the section 5. describing ibrutinib adverse events we added information concerning the so far identified mechanism contributing to ibrutinib intolerance. Changes were highlighted in green.
Reviewer 2 Report
Pula and his coworkers presented the mechanism of action of ibrutinib induce drug resistance and how to overcome it. The manuscript was well organized and written. The authors also introduced many compounds in the manuscript. I suggest the author should make a table for these compounds and summarize the mechanism of the action.
Author Response
Thank you for this suggestion. We added a table (Table 1) in the section 5 with short description and characterization of the most important, to our knowledge, compounds.
Reviewer 3 Report
1. The authors offered a well documented overview on the ibrutinib effects on chronic lymphocytic leukemia with regard to clinical efficacy as well as side effects and the settlement of cancer cell drug resistance. Alternative kinases inhibitors to this drug were also addressed within this review.
Nevertheless, the goal of the review must be underlined in the Introduction section and the authors should emphasize the major points regarding ibrutinib that were discussed in this review in comparison with other reviews on this topic.
2. Sections 8 and 9 must be supplemented with a short description of the reasons for the selection of drugs presented there (such as the main advantages over ibrutinib).
Author Response
Thank you for your comments. We modified the last paragraph of the introduction section to highlight the points and goal of this review. We added a table (Table 1) in the section 5 with short description and characterization of the most important, to our knowledge, compounds. The table includes also some of the compounds enlisted in sections 8 and 9. We reviewed the information given in these sections and information regarding the advantages and potential synergism of action is described in the text.